# Imaging the fate of histone Cse4 reveals *de novo* replacement in S phase and subsequent stable residence at centromeres

Jan Wisniewski[1,2]*, Bassam Hajj[1†], Jiji Chen[1], Gaku Mizuguchi[1,2], Hua Xiao[2], Debbie Wei[2], Maxime Dahan[1†], Carl Wu[1,2]*

[1]Janelia Farm Research Campus, Howard Hughes Medical Institute, Ashburn, United States; [2]Laboratory of Biochemistry and Molecular Biology, National Cancer Institute, National Institutes of Health, Bethesda, United States

**Abstract** The budding yeast centromere contains Cse4, a specialized histone H3 variant. Fluorescence pulse-chase analysis of an internally tagged Cse4 reveals that it is replaced with newly synthesized molecules in S phase, remaining stably associated with centromeres thereafter. In contrast, C-terminally-tagged Cse4 is functionally impaired, showing slow cell growth, cell lethality at elevated temperatures, and extra-centromeric nuclear accumulation. Recent studies using such strains gave conflicting findings regarding the centromeric abundance and cell cycle dynamics of Cse4. Our findings indicate that internally tagged Cse4 is a better reporter of the biology of this histone variant. Furthermore, the size of centromeric Cse4 clusters was precisely mapped with a new 3D-PALM method, revealing substantial compaction during anaphase. Cse4-specific chaperone Scm3 displays steady-state, stoichiometric co-localization with Cse4 at centromeres throughout the cell cycle, while undergoing exchange with a nuclear pool. These findings suggest that a stable Cse4 nucleosome is maintained by dynamic chaperone-in-residence Scm3.

**\*For correspondence:**
wisniewskij@janelia.hhmi.org (JW);
wuc@janelia.hhmi.org (CW)

**Present address:** †Laboratoire Physico-Chimie, CNRS UMR168, Institut Curie, Paris, France

**Competing interests:** The authors declare that no competing interests exist.

**Reviewing editor**: James T Kadonaga, University of California, San Diego, United States

## Introduction

In all eukaryotes, accurate segregation of genetic material constitutes the basis of cell division and inheritance. Chromosome segregation is controlled by a complex signalling network targeting the kinetochore—a protein superstructure of some 100 polypeptides, anchoring chromosomes to the mitotic spindle through interaction with a specialized region of the chromosome, the centromere. The chromatin structure of centromeres is distinguished from other chromosome regions by nucleosomes containing a distinct variant of histone H3, called CENP-A or CenH3 (*Biggins, 2013*; *Westhorpe and Straight, 2013*).

Unlike other organisms, in which centromeres encompass extended regions with tens or thousands of CENP-A nucleosomes, the centromere of the budding yeast *Saccharomyces cerevisiae* is fully specified by a short DNA segment (*CEN*, ~125 bp) (*Gaudet and Fitzgerald–Hayes, 1987*; *Murphy et al., 1991*). This so-called 'point' centromere consists of a single nucleosome-like chromatin particle containing Cse4, the yeast ortholog of CENP-A (*Stoler et al., 1995*; *Meluh et al., 1998*). Classic genetic, molecular, and biochemical studies have defined three contiguous centromeric DNA elements CDEI, CDE II, and CDE III that direct assembly of a Cse4 nucleosome by sequence-specific DNA binding factors CBF1 and CBF3 (*Cai and Davis, 1990*; *Lechner and Carbon, 1991*) and Scm3, a Cse4-specific chaperone (*Camahort et al., 2007*; *Mizuguchi et al., 2007*; *Stoler et al., 2007*; *Xiao et al., 2011*; *Cho and Harrison, 2011b*). The singular nature of centromeric nucleosomes of budding yeast thus

**eLife digest** When cells multiply, it is essential for each new cell to get a copy of the organism's genetic blueprint. If an error occurs during cell division, and one of the daughter cells ends up with too many or too few copies of a chromosome, the cell can die or malfunction. Errors during cell division can, for example, cause cancer.

Before a cell divides, it must create an exact copy of each of its chromosomes. The two copies of the chromosome are linked together at a region called the centromere. To separate them, structures called microtubules attach to each side of the centromere via a structure called the kinetochore. The kinetochore then sends out signals orchestrating how the microtubules should move in order to pull the chromosomes apart.

In yeast, it is known that a protein called Cse4 must be present at the centromere for cell division to be successful. However, researchers have come to conflicting conclusions about how many copies of this protein are needed and how they function as the chromosome copies are separated.

Wisniewski et al. now reveal that a 'tag' scientists use to make Cse4 more visible under a microscope may have skewed the results of some studies. Attaching a tag to the end of the protein interferes with its function, slowing down cell growth, and even killing cells at high temperatures. This could explain the disagreements about how Cse4 works.

Placing a tag inside Cse4, on the other hand, allows the protein to behave normally. Using such an internal tag, Wisniewski et al. found that, as the cell copies its chromosomes, old Cse4 is removed and replaced by new molecules. Those proteins then remain attached to the centromere throughout cell division. A second protein called Scm3 helps to hold the Cse4 in place.

By clarifying the number and behavior of various crucial components of the kinetochore, this work opens avenues to better understand the process of chromosome separation.

offers a simplified biological system for detailed study of the biogenesis, maintenance, and dynamics of centromere–kinetochore interactions.

Despite this simplicity, the architecture of Cse4 nucleosomes has become the subject of much debate. Cse4 nucleosomes have been reported to differ from the canonical nucleosome not only by the replacement of both molecules of histone H3 by the Cse4 variant, but also by the presence of chaperone Scm3 and dislocation of histones H2A-H2B (*Mizuguchi et al., 2007*; *Xiao et al., 2011*), or by existence of a hemisome particle bearing half the histone content (*Dalal et al., 2007*; *Furuyama et al., 2013*). Moreover, live cell microscopy of GFP-tagged Cse4 have variously indicated that the number of Cse4 molecules associated with centromeres may be either several-fold greater than the two Cse4 molecules within a nucleosome (*Coffman et al., 2011*; *Lawrimore et al., 2011*), or oscillate during mitosis from one to two molecules per centromeric nucleosome (*Shivaraju et al., 2012*). Thus, the fundamental composition and stability of the Cse4 nucleosome has been obfuscated by the recent microscopic studies.

To assess those claims, we have taken a direct approach to monitor the fate of Cse4 molecules throughout the cell cycle in live yeast. We utilize the photoconvertible fluorescent protein tdEos in fluorescence pulse-chase experiments to mark pre-existing Cse4 and document its complete replacement at centromeres with newly synthesized molecules early in S phase. We find that after this transient replacement, Cse4 remains stably associated with centromeres for the rest of the cell cycle, without additional Cse4 deposition in anaphase. Importantly, we show that recent discrepant claims can be attributed to reliance on GFP fusion to the C-terminus of Cse4, which causes impaired cell growth, temperature-dependent lethality, and extra-centromeric nuclear accumulation. By contrast, an insertion of GFP or tdEOS within the unstructured N-terminal tail of Cse4 avoids such deleterious phenotypes. Hence, many of the conflicting properties of C-terminally tagged Cse4 reflect the behavior of functionally impaired protein rather than native Cse4.

## Results

### Internal tag reveals exclusive centromeric localization of Cse4

To analyze the localization of Cse4 in live cells, we introduced a fluorescent protein tag at an internal Xba I site (corresponding to Leu81 within the long N-terminal tail of Cse4) based on the original

studies of *Stoler et al. (1995)* and *Chen et al. (2000)* (*Figure 1A*). These workers showed that insertions or deletions within the N-terminal tail do not impose any deleterious growth phenotype at all tested temperatures, as long as a 33-residue essential END domain, that interacts with the Ctf19-Mcm21-Okp1 kinetochore sub-complex, is preserved. Thus, as schematically depicted in *Figure 1B*, the flexible N-terminus of Cse4 is well suited to accommodate internal protein tags. In contrast, the extreme C-terminal residues of Cse4 (QFI, aa 227-229, located close to the structured part of the nucleosome [*Tachiwana et al., 2011*]) mediate recognition by CENP-C (*Kato et al., 2013*) and an adjoining tag is likely to impair this interaction. Moreover, functionality of *Drosophila* CENP-A/CenH3 is also preserved by an internal insertion of GFP but not by a C-terminal fusion (*Schuh et al., 2007*).

Accordingly, we investigated the behavior of Cse4 internally tagged with GFP (*Cormack et al., 1997*) or the photoconvertible fluorophore tdEos (tandem dimer Eos; *Nienhaus et al., 2006*). For both

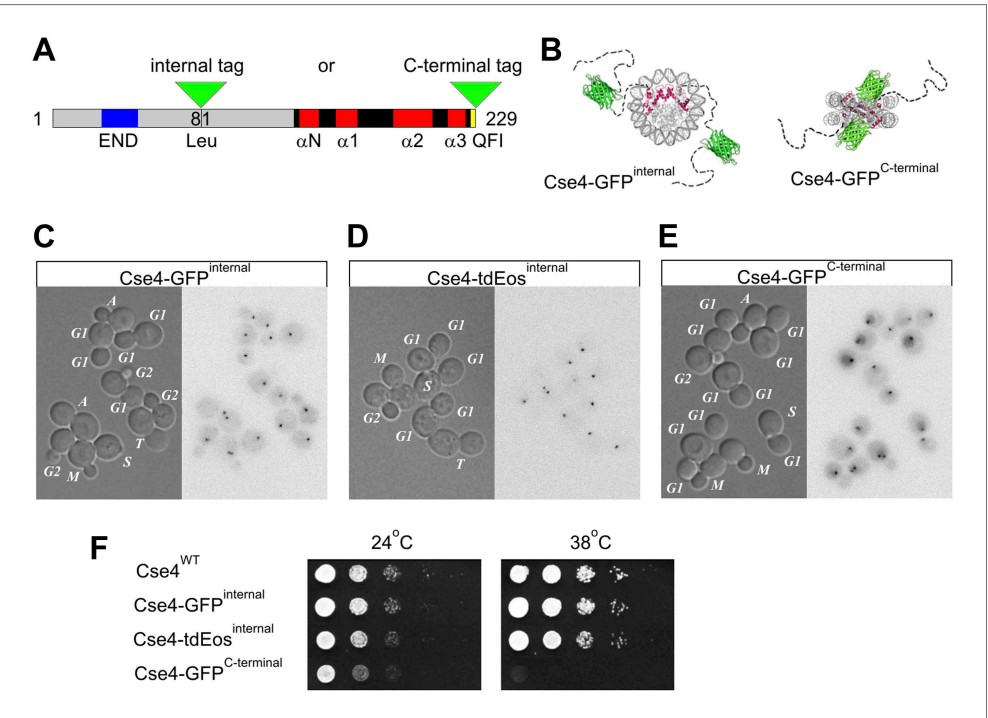

**Figure 1**. Internal tagging of Cse4 confers exclusive centromeric localization and preserves wild type phenotype. (**A**) Alternative tag locations at Leu81 (internal XbaI site) or at the C-terminus of Cse4 are indicated by green triangles. Unstructured N-terminal tail (aa1-135) is depicted in grey while region corresponding to the known 3D structure of mammalian CENP-A (aa134-226) is shown as solid black and red (loops and α-helices of histone-fold domain). Functionally important END region (aa28-60) and C-terminal CENP–C interaction region QFI (aa227-228) are highlighted in blue and yellow. (**B**) Schematic position of fluorescent protein tags in relation to the overall nucleosome structure. Monomeric GFP tag is shown in green while Cse4 histone-fold domains are highlighted in red inside nucleosome core. Unstructured N-terminal tails of Cse4 are depicted as dashed lines for illustrative purposes. (**C**–**E**) Distribution of tagged Cse4 in live cells containing Cse4 tagged internally with GFP (**C**) or tdEos (**D**), or the C-terminal GFP fusion (**E**). Cell cycle stages are indicated in DIC panels. In addition to G1, S, and G2, individual stages of mitosis are identified as: M—metaphase, A—anaphase, T—telophase. Fluorescence images are shown as negatives to reveal residual intracellular autofluorescence and the diffuse nuclear presence of C-terminally tagged Cse4. (**F**) Viability test of strains containing wild-type or tagged Cse4. 10 μl of 10-fold serial dilutions of equivalent log-phase cultures were spotted on YPD plates and incubated overnight at 38°C or for 36 hr at 24°C.

The following figure supplements are available for figure 1:

**Figure supplement 1**. Fluorescence of centromeric clusters containing C-terminally tagged Cse4 is slightly elevated but does not double in anaphase.

**Figure supplement 2**. Internally tagged Cse4 accumulates at levels comparable to wild-type protein.

constructs, we replaced the wild-type *CSE4* gene in a haploid yeast strain, yielding tagged strains displaying normal bud morphology (*Figure 1C,D*), and viability at normal and elevated growth temperatures indistinguishable from the wild-type strain (*Figure 1F*).

Live cell imaging of internally tagged Cse4-GFP reveals fluorescence exclusively in a single dot or a pair of dots (*Figure 1C*), corresponding to the clusters of yeast centromeres (*Jin et al., 1998*; *Meluh et al., 1998*; *Chen et al., 2000*; *Jin et al., 2000*). Identical results are obtained for internal Cse4-tdEos fusion, despite the larger tag size (*Figure 1D*). Additionally, the red emission of photoconverted tdEos avoids intracellular autofluorescence and improves contrast–nonetheless, no nuclear fluorescence is detectable outside of centromeric clusters. Curiously, S phase centromeres display weaker tdEos fluorescence (*Figure 1D*), a phenomenon further explored below.

## C-terminal tag leads to extra-centromeric Cse4 accumulation and impaired viability

Live cell imaging studies relying on a GFP fusion to the C-terminus of Cse4 reported unusual properties of Cse4 (*Coffman et al., 2011*; *Lawrimore et al., 2011*; *Shivaraju et al., 2012*). Therefore, for comparison with our internal GFP fusions, we examined a representative C-terminally tagged Cse4-GFP strain (MSY173, obtained from Jennifer Gerton's laboratory; *Shivaraju et al., 2012*). For this strain, we confirm the presence of fluorescent centromeric clusters, though centromere intensity appears slightly elevated (*Figure 1—figure supplement 1*). However, in contrast to internally tagged Cse4, we clearly detect extra-centromeric fluorescence throughout nuclei at every stage of the cell cycle (*Figure 1E*, *Figure 1—figure supplement 1*). This difference is confirmed by Western blot analysis of whole cell extracts, showing ~twofold excess of C-terminal over internal Cse4-GFP fusion of comparable size and blotting efficiency (*Figure 1—figure supplement 2A*), while internally tagged Cse4 is present at levels close to wild-type Cse4 (*Figure 1—figure supplement 2B–G*). Most importantly, the strain carrying the C-terminal tag shows substantially reduced viability. The C-terminal Cse4-GFP strain exhibits slow growth in rich medium even at 24°C, and is not viable at 38°C, while none of the internal fusions have growth defects at either temperature (*Figure 1F*). Taken together, our results demonstrate that fusion of a fluorescent protein tag to the C-terminus impairs Cse4 function. Accordingly, we only used internal tags to further explore the physiological dynamics of Cse4.

## Pulse-chase shows replacement of Cse4-tdEos at entry into S phase

After synthesis and protein folding, tdEos fluorophores undergo relatively slow maturation to a green fluorescent state (*Nienhaus et al., 2006*) (*Figure 2A*, *Figure 6—figure supplement 1*). However, upon exposure to violet light, such mature fluorophores undergo almost instantaneous, irreversible photoconversion to a red-emitting state. To follow the fate of Cse4-tdEos in living cells by fluorescence pulse-chase analysis (*Figure 2B*), we photoconverted Cse4-tdEos in asynchronously growing yeast to mark its initial distribution at different cell cycle stages (*Figure 2C*). This reveals centromeric clusters in all cells, including the aforementioned weak signal in early S phase. Cells are then allowed to advance into the cell cycle, and re-imaged 40 min later. *Figure 2D* shows that centromeric clusters typically retain pre-existing Cse4, with the striking exception of cells crossing the G1/S boundary (magenta outlines), which lose centromeric fluorescence. This indicates that pre-existing Cse4 is not maintained or recycled in S phase. An additional round of photoconversion at the end of the experiment confirms loading of new Cse4 molecules at centromeric clusters (*Figure 2E*), in accordance with previous studies showing Cse4-GFP deposition in S phase (*Pearson et al., 2004*; JW, personal communication).

*Figure 2F* shows a specific example of S phase replacement of Cse4. A cell in telophase displays equivalent Cse4-tdEos fluorescence on both centromere clusters. Thereafter, the mother cell, which enters S phase sooner than the daughter, loses pre-existing centromeric signal, whereas centromeres of the daughter cell, still in G1, are still occupied by pre-existing Cse4. A second round of photoconversion confirms that mother cell centromeres contain newly deposited Cse4. By contrast, photoconversion of a cell in S phase reveals weak fluorescence of the centromeric cluster (*Figure 2G*). Upon advancement to telophase, the original cluster separates into two, each still showing weak fluorescence. A second round of photoconversion reveals a substantial increase of signal at these telophase clusters, a phenomenon attributable to the maturation of the tdEos fluorophore, as shown below.

Taken together, our results document that early in S phase, Cse4 molecules are eliminated and replaced by newly synthesized molecules, thereafter remaining stably associated with centromeres

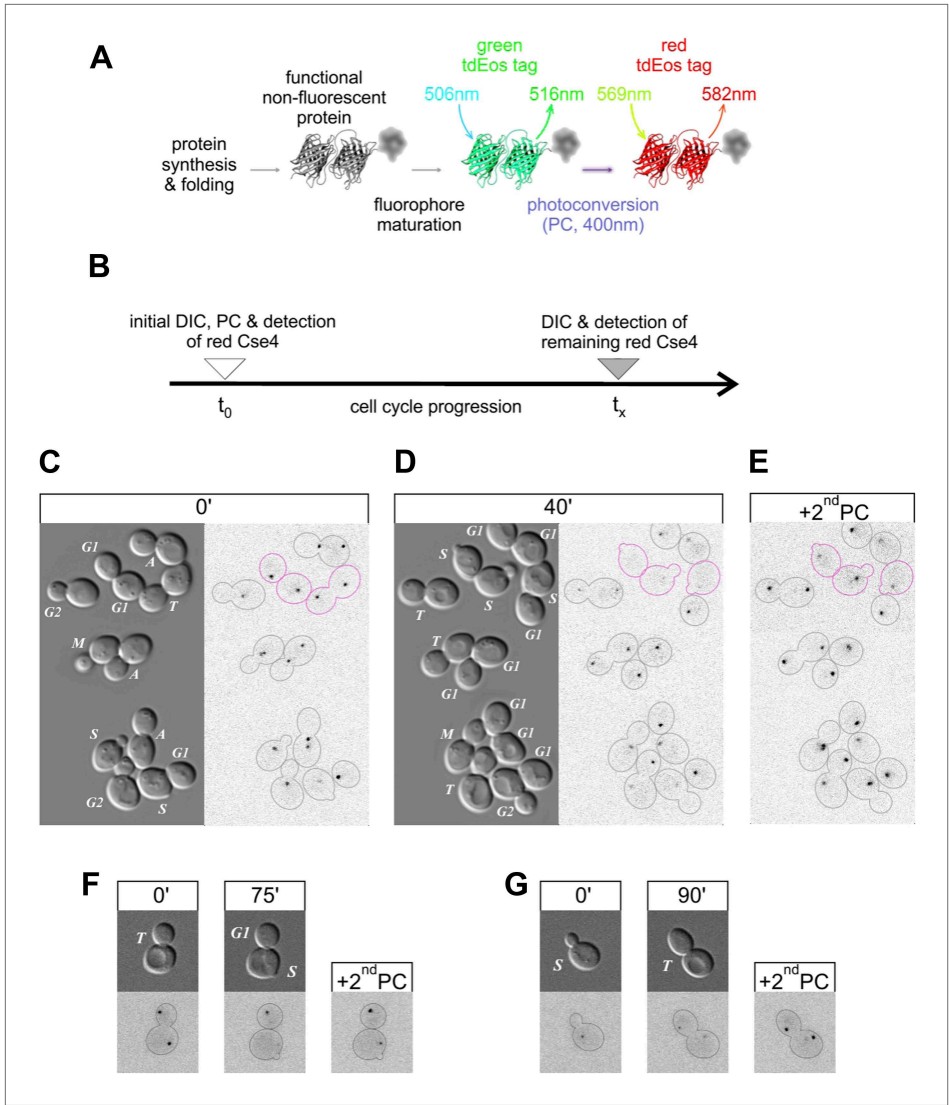

**Figure 2**. Pre-existing Cse4 is removed and exchanged for new Cse4 molecules at G1/S transition. (**A**) Relevant fluorescence states of a tdEos-tagged protein molecule are depicted schematically after its synthesis and folding, fluorophore maturation and irreversible photoconversion. Excitation and emission peak wavelengths are indicated (see *Figure 6—figure supplement 1* for additional details). (**B**) Pulse-chase experimental scheme. After initial photoconversion (pulse at $t_0$), red-fluorescent Cse4 is followed into later stages of the cell cycle (chase until $t_x$). (**C–E**) Cells containing Cse4-tdEos were imaged immediately after pulse (**C**) and following 40 min chase (**D**). At the end, additional photoconversion (2ndPC) was used to confirm sufficient Z-stack range (**E**). Three cells that crossed G1/S boundary are outlined in magenta while all other cells are outlined in grey, based on DIC images. (**F**) An example of a telophase cell followed until mother cell entered S phase, while the bud-derived daughter remained in G1. (**G**) An example of S phase cell followed into telophase.

through the rest of the cell cycle. The results also indicate the absence of a persistent pool of free nuclear or cytoplasmic Cse4, as we fail to observe carry-through of pre-existing Cse4 into S phase.

## Fluorophore maturation accounts for fluorescence increase after deposition

The fluorescence of Cse4 clusters increases gradually from S phase through mitosis for both tdEos and GFP insertions (*Figure 3A*). Such a pattern could be caused by either a continuous deposition of newly synthesized fluorescent molecules (precluded by results above), or an ongoing maturation of fluorophores already deposited in S phase. We tested the second scenario by measuring the fluorescence of

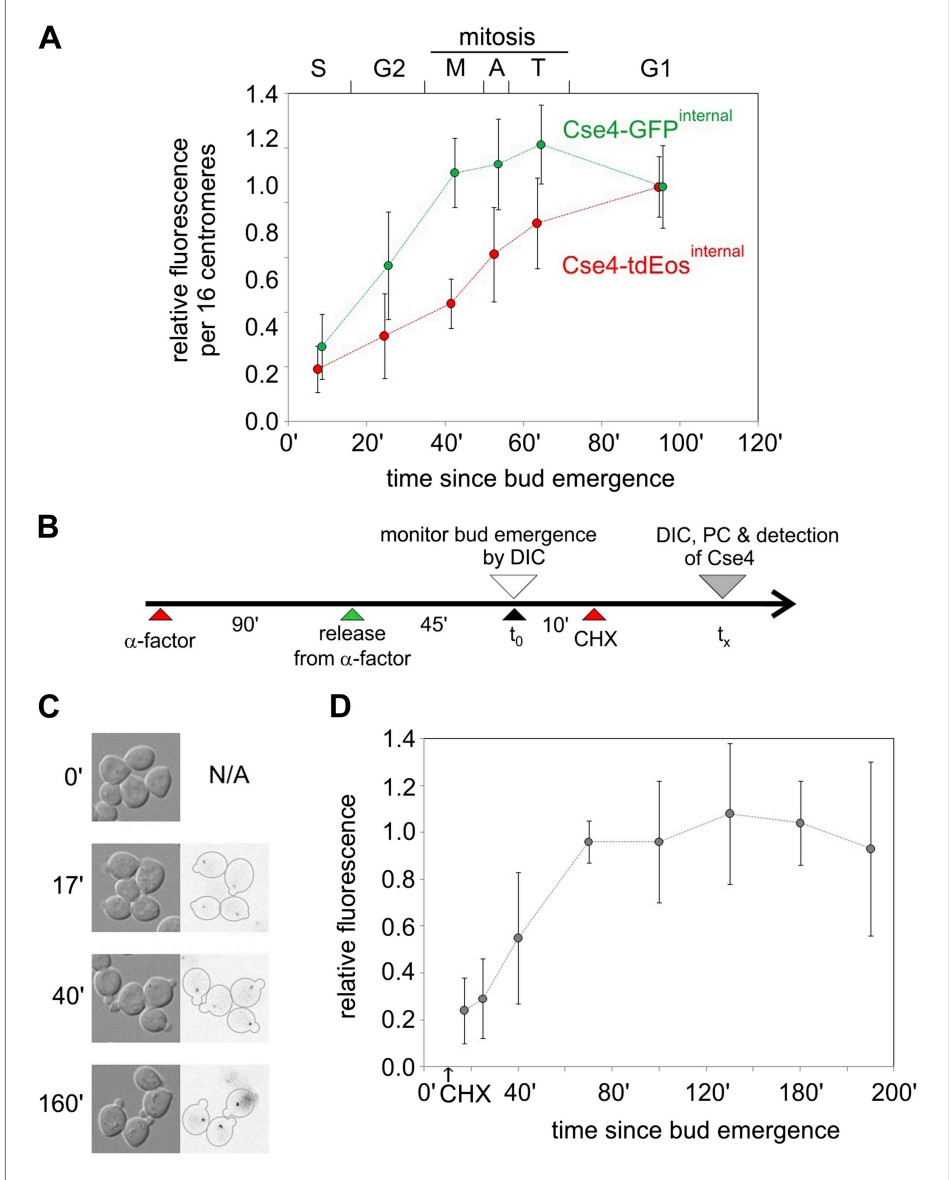

**Figure 3**. Cell cycle-dependent increase in centromere cluster intensity is a result of fluorophore maturation. (**A**) Relative intensity of centromeric clusters in asynchronously growing cells as the function of cell cycle stage and approximate time since entry into S phase. Values were corrected per 16 centromeres, to account for the presence of replicated (32) centromeres in a single 'dot' in S and G2. ~7500 and ~11,000 photons were detected during 5 s exposure for G1 clusters containing Cse4 with internal GFP or tdEos, respectively—other results were normalized against those values. Standard deviation of each sample is indicated. (**B**) Schematic of the experiment to measure maturation rate of fluorophores present on Cse4-tdEos. Following α-factor synchronization, cycloheximide (CHX, 0.2 mg/ml) was added 10 min after entry into S phase. (**C**) Images of S phase cells at different sampling points ($t_x$) after bud emergence. Cells which entered S phase prior to addition of cycloheximide are outlined. (**D**) Relative fluorescence of Cse4-tdEos centromere clusters in S phase in the absence of protein synthesis. Plateau value (average of points representing $t_x$ >100 min, ~1750 photons detected during 1 s exposure) was used for normalization and standard deviations are indicated.

newly formed centromere clusters after protein synthesis was blocked with cycloheximide (**Figure 3B**). We find that their brightness increases with time until it reaches a plateau at ~70 min, remaining stable for at least 120 min thereafter, notwithstanding the cycloheximide-induced block in cell cycle progression (**Figure 3C,D**). This intensity profile indicates that the half-time of maturation at 25°C is

approximately 40 min—similar to the time observed for half-maximal increase of Cse4-tdEos fluorescence in a population of asynchronously growing cells (*Figure 3A*). Hence, fluorophore maturation is sufficient to explain the gradual rise of Cse4-tdEos fluorescence through the cell cycle after early S phase deposition.

## Cse4 replacement correlates with DNA replication

To gain insight into the mechanism of Cse4 replacement, we investigated the role of DNA replication by analysis of synchronized cells in which replication is blocked with hydroxyurea (*Figure 4A*). We photoconverted Cse4-tdEos in late G1, ~30 min after release from α-factor arrest (corresponding to ~15 min prior to bud emergence), and followed the fate of the pre-existing Cse4 thereafter. *Figure 4B* shows that, as expected, control untreated cells lose strong centromeric fluorescence upon entry into S phase. However, cells treated with hydroxyurea uniformly retain pre-existing Cse4 on centromere clusters, regardless of bud emergence. This suggests that the removal of old Cse4 from centromeres is associated with DNA replication.

## No additional deposition of Cse4 on centromeres in anaphase

The gradual increase of Cse4 fluorescence through the cell cycle conflicts with the discrete twofold increase reported for C-terminally tagged Cse4-GFP at anaphase (*Shivaraju et al., 2012*). To further examine this issue, we used a targeted FRAP procedure to detect any deposition of internally tagged Cse4-tdEos at anaphase (*Figure 5A*). We photoconverted Cse4-tdEos in a metaphase cell to reveal the pair of centromere clusters (*Figure 5B1*). One of those clusters was then photobleached with a pulsed dye laser beam focused to a diffraction-limited spot, without affecting fluorescence of the other cluster (*Figure 5B2*). Upon progression through anaphase, we find that only one red-fluorescent cluster is visible at telophase as well (*Figure 5B3*). This indicates that additional Cse4 deposition did not occur on the bleached cluster, nor did Cse4 exchange between the two clusters. A second photoconversion conducted at the end of the experiment (uncovering additional fluorophores that completed maturation in the meantime) confirms that the targeted cluster remains functional and segregates to the opposite pole (*Figure 5B4*). Hence, our results indicate a compositional stasis for Cse4 after S phase deposition, and do not support a second wave of Cse4 deposition in anaphase.

## Centromeric cluster size and compaction in anaphase

Recent advances in fluorescence microscopy enable localization of molecules in live and fixed cells with sub-diffraction accuracy (*Sengupta et al., 2012*). The newly developed multifocal microscope (MFM) allows 3D imaging of the entire yeast cell volume in a single exposure (*Abrahamsson et al., 2013*) and, when combined with PALM (*Betzig et al., 2006*), permits super-resolution localization of single fluorescent molecules with lateral accuracy of ~20 nm and axial accuracy of ~50 nm within a depth of ~4 µm (Hajj et al., unpublished data). We applied this combined approach to analyze the volumetric distribution of Cse4-tdEos molecules within centromeric clusters in paraformaldehyde-fixed cells.

As illustrated in *Figure 6A*, individual tdEos fluorophores are detectable simultaneously at different depths within a fixed cell, and a low photoconversion rate ensures observation of well-separated single-molecule fluorescence events (*Figure 6B*). A resulting plot of the 3D distribution

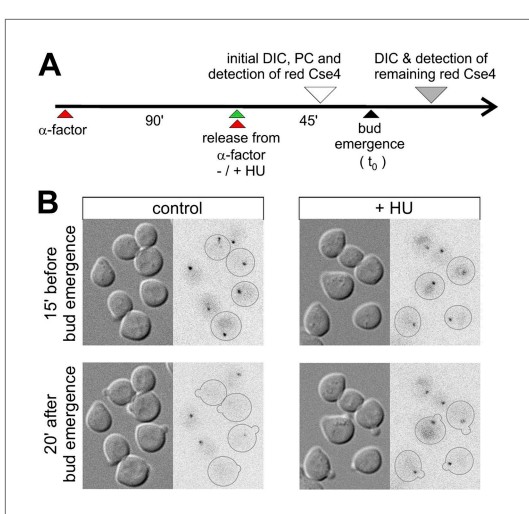

**Figure 4**. Removal of pre-existing Cse4 is associated with DNA replication. (**A**) Experimental scheme to assess role of DNA replication on the removal of pre-existing Cse4. α-factor synchronized cells were released into control medium or one with 0.2 M hydroxyurea (HU). Cse4-tdEos was photoconverted prior to bud emergence and then followed after sizable buds became evident. (**B**) Examples of cells released from α-factor block directly into control or hydroxyurea (+HU) containing medium. Time of photoconversion and observation after chase is indicated. Only cells on which buds appeared during the observation period are outlined.

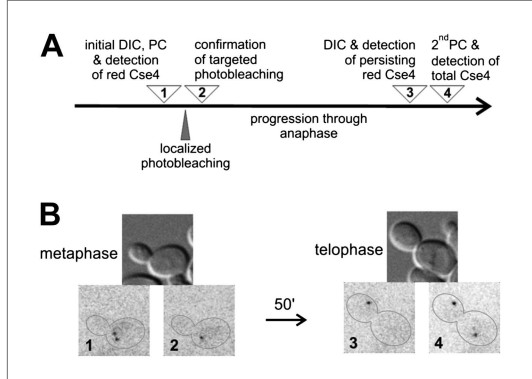

**Figure 5**. There is no additional Cse4 deposition in anaphase. (**A**) Scheme of the experimental test for Cse4 deposition in anaphase. All operations were carried on a selected metaphase cell in the specified order. 3D diffraction-limited spot, generated with a galvano-controlled MicroPoint 551 nm dye laser system, was used for targeted photobleaching. (**B**) An example of the metaphase cell subjected to a targeted photo-bleaching of photoconverted Cse4-tdEos centromeric cluster. Images were acquired at stages indicated in panel **A**.

of all independent detections inside an anaphase cell (assembled with ViSP software; *El Beheiry and Dahan, 2013*) reveals both centromere clusters as compact groups of 20 and 22 tdEos fluorophores (*Figure 6C*; *Video 1*). These should not be construed to reflect the total number of Cse4 molecules present at the clusters, because incomplete maturation and the initial photobleaching prior to PALM (necessitated by paraformaldehyde-induced conversion—JW, personal communication) leave only a fraction of total fluorophores detectable as single-molecule events. Moreover, the existence of the reversible dark state of red tdEos may cause multiple detections of some fluorophores (*Annibale et al., 2010*; *Lee et al., 2012*; *Figure 6—figure supplement 1*). Despite this, MFM-PALM localization of individual fluorophores allows estimation of the overall dimensions of centromere clusters. We find that Cse4 clusters in G1 are typically ~450 nm across (*Figure 6D,F*; *Video 2*), clearly indicating that their wide-field image (the sum of all individual Airy disks) would significantly exceed the diffraction limit (in this case an Airy disk with FWHM ~225 nm). Strikingly, anaphase clusters are more compact and asymmetric, on average approximating an ellipsoid of 350 nm × 200 nm (still above the diffraction limit—*Figure 6E,G*; *Video 3*). This change corresponds to ~threefold reduction in the volume of the cluster and thus higher spatial density of centromeres. Frequently, the shortened polar axis coincides with the direction of the mitotic spindle extending between anaphase clusters (*Video 1*). Such substantial dimensions of Cse4 clusters and their compaction in anaphase have important implications for photometric measurements of fluorescence intensity (see below).

In addition to centromere clusters, we also observe individual fluorescent events scattered throughout the cytoplasm. Due to the absence of a persistent free Cse4 pool (as demonstrated by pulse-chase experiments in *Figure 2*), those are unlikely to represent free Cse4 molecules. Because GFP is known to be resistant to proteolytic degradation (*Chiang et al., 2001*), we speculate that these cytoplasmic events correspond to fluorophore moieties persisting after proteolytic degradation of unincorporated Cse4 (*Collins et al., 2004*). Such residual fluorophores would not be distinctly detectable in live cells due to their mobility and dispersal in the cytoplasmic volume (~40-fold larger than the nucleus).

## Two Cse4 molecules are present at each centromere

To estimate the number of Cse4-GFP molecules present at a centromere cluster, we compared its fluorescence intensity to that of TetR-GFP bound to a defined number of tet operator sites (tetO) (*Michaelis et al., 1997*). To minimize the background caused by free TetR-GFP molecules, we expressed TetR-GFP from a weakened, non-induced URA3 promoter (*Roy et al., 1990*).

*Figure 7A* shows that a fluorescent dot is detectable against a diffuse nuclear background even in cells containing 7x tetO and becomes clearly apparent in the case of 14x tetO. When compared at an identical brightness scale, it is evident that the intensity of Cse4-GFP cluster lies between that of 28 and 42 GFPs, the maximum number that can be present on 14x and 21x tetO, respectively, as tetracycline repressor is a homodimer. Furthermore, we performed photometric measurements of tetO arrays and centromeric clusters after precise background subtraction. We utilized wavelet filtering (*Berry and Burnell, 2011*) to separate small scale features (e.g., clusters) from larger patterns (e.g., nuclear and cytoplasmic fluorescence) (*Figure 7—figure supplement 1*). *Figure 7B* shows that median intensity of wavelet filtered Cse4-GFP clusters corresponds to ~36 GFP molecules. Given the scatter of measured values, this is consistent with two Cse4 molecules for each of the 16 centromeres clustered together in telophase.

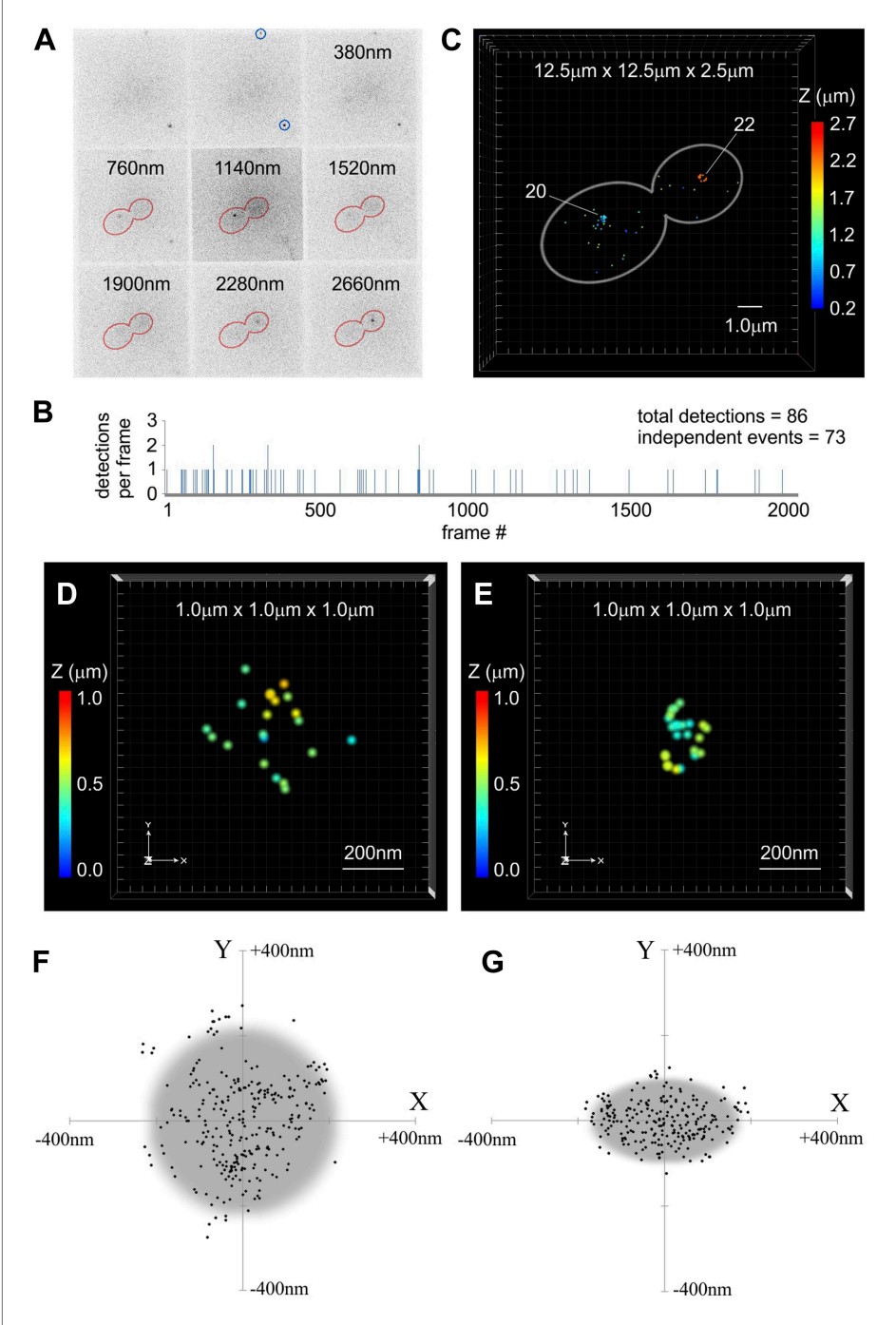

**Figure 6**. Centromeric clusters become more compact during anaphase. (**A**) An example of MFM-PALM image with nine simultaneously acquired Z-planes. Gold Nanorods are indicated (blue circles) and distance above the glass surface is listed for other tiles. Anaphase cell outline (red) is based on a separate bright-field MFM image. Two single-molecule events of Cse4-tdEos are visible inside the cell. (**B**) A time-trace representation of the number of tdEos fluorophore detections per frame during single MFM-PALM acquisition series. Events lasting >1 frame were considered to represent the same fluorophore. (**C**) A projection of all 73 independent single-fluorophore detections in the above image series. Image volume is indicated and Z position of individual localizations is color-coded. Each event is depicted as a dot 50 nm across (increased from average 20 nm lateral precision to facilitate visualization at this image scale). Number of events concentrated at each of the two centromeric clusters is listed. See **Video 1**. ViSP software (**El Beheiry and Dahan, 2013**) was used for projections in **C**–**E**. (**D**) A representative example of a 3D

*Figure 6. Continued on next page*

*Figure 6. Continued*

distribution of Cse4-tdEos molecules on the G1 centromere cluster. Each event (total of 21 independent detections) is depicted as a 20 nm dot, corresponding to the average lateral localization precision. Total volume of 1 μm$^3$ is shown, with color-coded Z distance. See *Video 2*. (**E**) A representative late anaphase centromere cluster depicted as above. Total of 24 independent detections are plotted. See *Video 3*. (**F**) Compilation of detections from 10 G1 clusters, center-aligned and projected onto XY plane. Grey circle depicts cross-section of a sphere (~450 nm across) sufficient to contain majority of detected Cse4-tdEos molecules. (**G**) Compilation of total detections from 10 late anaphase and telophase clusters projected onto XY plane (center-aligned, long axis rotated horizontally). Grey ellipse depicts cross-section of an ellipsoid (~350 nm equatorial diameter and ~200 nm polar distance) sufficient to contain majority of detected Cse4-tdEos molecules. In both cases, distribution in Z is comparable to that along X-axis (not shown).

The following figure supplements are available for figure 6:

**Figure supplement 1**. tdEos fluorophore undergoes transitions between multiple fluorescent and dark states.

## Steady-state centromeric occupancy and dynamic exchange of Scm3

The high stability of centromeric Cse4 nucleosomes after S phase suggests a need for special maintenance mechanism(s). Scm3 is the Cse4-specific chaperone required for Cse4 deposition and maintenance at centromeres (*Camahort et al., 2007*; *Mizuguchi et al., 2007*; *Stoler et al., 2007*). Scm3 is recruited to centromeres by sequence-specific factor CBF3 and itself possesses AT-rich DNA binding activity (*Xiao et al., 2011*; *Cho and Harrison, 2011a*). We demonstrated previously that Scm3-GFP localizes to centromeres at every stage of the cell cycle, including anaphase of mitosis, and is also distributed diffusely throughout the nucleus (*Xiao et al., 2011*); Scm3-tdEos has an identical distribution (*Figure 8—figure supplement 1A*). Comparison of total nuclear and centromeric fluorescence reveals ~fourfold excess of free Scm3-tdEos throughout the nucleus compared to the centromeric-bound protein (*Figure 8—figure supplement 1B*).

To assess the stability of centromeric Scm3, we applied targeted laser photobleaching of Scm3-tdEos at one of two centromere clusters (*Figure 8A*). In contrast to the stability of Cse4-tdEos, Scm3 fluorescence reappears on the cluster within several minutes (between 4 and 11 min in this example), demonstrating that, unlike Cse4, Scm3 undergoes exchange between centromeres and the free nuclear pool. When measured throughout the cell cycle, the average recovery time (at which fluorescence is detected again) is ~5 min (*Table 1*). Such dynamic exchange of Scm3 with persistent, steady-state occupancy may ensure continuing integrity of the singular centromeric nucleosome after Cse4 deposition in S phase.

A pulse-chase experiment shows that total pre-existing Scm3 persists through multiple cell cycles, with gradual dilution during consecutive cell divisions, indicating its low rate of turn-over (*Figure 8B*). Moreover, steady-state fluorescence of centromeric Scm3 shows a mild decrease in S phase, suggesting synthesis of new Scm3 molecules with immature fluorophores during that stage (*Figure 8—figure supplement 1C*). Comparison of centromeric Scm3 and Cse4 fluorescence in telophase, when the majority of tdEos fluorophores are mature, shows that their intensities closely overlap and follow similar photobleaching curves (*Figure 8C*). An identical result is also obtained with the GFP tag (*Figure 8—figure supplement 2*). Taking into account that Cse4 remains stable after deposition and Scm3 interacts dynamically and persistently with centromeres, this indicates that near equimolar levels of both proteins coexist on centromere clusters throughout the cell cycle.

**Video 1**. 3D representation of tdEos fluorophore distribution in anaphase cell from *Figure 6C*. Each event is depicted as a dot 50 nm across instead of the actual average localization precision (20 nm lateral/50 nm axial). The original color coding of axial distance from *Figure 6C* is maintained. The video was assembled in ViSP software (*El Beheiry and Dahan, 2013*).

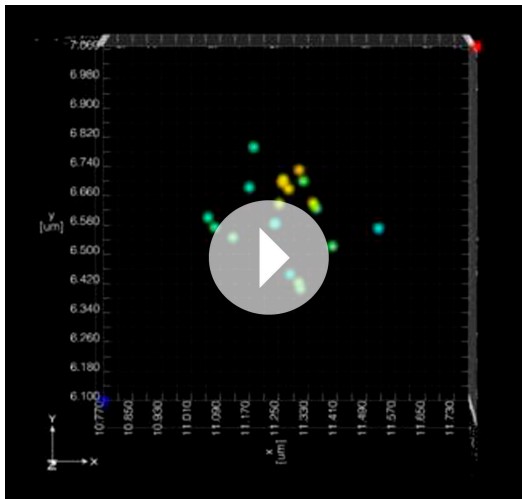

**Video 2**. 3D representation of Cse4-tdEos distribution in G1 centromere cluster from **Figure 6D**. Each event is depicted as a dot 20 nm across instead of the actual average localization precision (20 nm lateral/50 nm axial). The box encloses 1 μm³ volume and two artificial sizing marks (red and blue) are present at the corners. The original color coding of axial distance (**Figure 6D**) is maintained. The video was assembled in ViSP software (**El Beheiry and Dahan, 2013**).

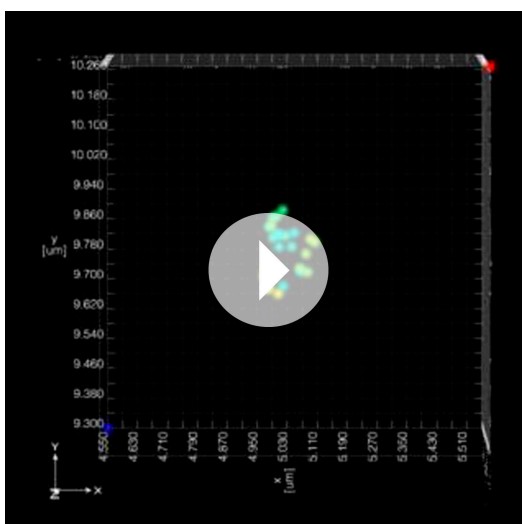

**Video 3**. 3D representation of Cse4-tdEos distribution in late anaphase centromere cluster from **Figure 6E**. Each event is depicted as a dot 20 nm across instead of the actual average localization precision (20 nm lateral/50 nm axial). The box encloses 1 μm³ volume and two artificial sizing marks (red and blue) are present at the corners. The original color coding of axial distance (**Figure 6E**) is maintained. The movie was assembled in ViSP software (**El Beheiry and Dahan, 2013**).

## Discussion

In a side-by-side comparison, we document that a C-terminal GFP tag impairs Cse4 functionality, causing severe growth defects and substantial extra-centromeric accumulation. We observed similar growth defects with a FLAG epitope tag as well (GM, unpublished data). The extreme C-terminal residues of Cse4 specify recognition by Mif2, the yeast CENP-C inner kinetochore protein (**Kato et al., 2013**). Accordingly, a C-terminal fusion is likely to affect such interaction, perturbing kinetochore functionality. Thus, the molecular phenotypes of C-terminally tagged Cse4-GFP reflect properties of functionally impaired Cse4, rather than the native protein. Similarly, partial loss of function was also observed for C-terminally tagged CENP-A/CenH3 in mouse (**Kalitsis et al., 2003**) and Drosophila (**Schuh et al., 2007**).

Recent claims of altered cell cycle dynamics and/or substantially increased centromere localization were based on such compromised Cse4 fusions, despite their temperature-sensitive phenotype and evident extra-centromeric distribution (**Coffman et al., 2011**; **Lawrimore et al., 2011**; **Shivaraju et al., 2012**). On the other hand, the very first epitope tag in Cse4 consisted of an insertion within the N-terminal tail, at codon 81 (**Stoler et al., 1995**). At this location, a GFP tag does not affect Cse4 functionality and cell growth (**Chen et al., 2000**). We find that even the insertion of tdEos tag (twice the size of GFP) at the same location is well tolerated, causing no detectable growth phenotypes, and similar findings were obtained for up to four GFP copies (R Baker, personal communication). Thus, internally tagged Cse4 fusions should be used in imaging studies as a preferred reporter of the composition and dynamics of centromeric nucleosomes.

The internal photoconvertible tdEos tag allows a direct analysis of Cse4 dynamics in live cells, minimizes autofluorescence, improves signal to noise, and enables excitation at low energies to limit phototoxicity and cell cycle perturbation. This reveals replacement of Cse4 exclusively in early S phase, linked to DNA synthesis—consistent with the timing of centromere replication (**McCarroll and Fangman, 1988**; **Pohl et al., 2012**). Our data elaborate on the S phase deposition of Cse4 reported by **Pearson et al. (2004)**, by showing this process as a removal of pre-existing Cse4 followed by the deposition of newly synthesized molecules, without recycling of old Cse4. Subsequently, Cse4 remains stably bound to centromeres for the remainder of the cell cycle until the next S phase. Furthermore, targeted photobleaching experiments show no second wave of

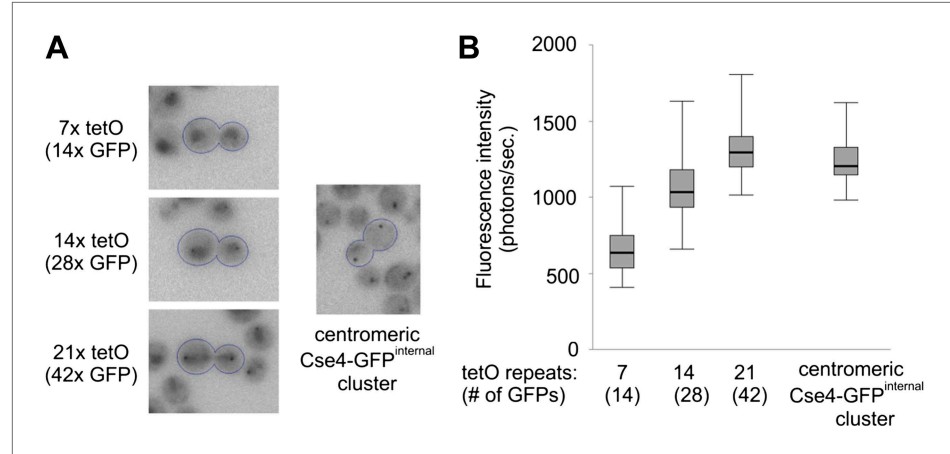

**Figure 7**. Two Cse4 molecules are present on each centromere. (**A**) Comparison of Cse4-GFP centromere clusters with TetR-GFP bound to arrays of 7, 14 or 21 tetO, displayed within the same brightness range. Representative telophase cells are outlined. Clusters in surrounding cells may be out of focus. (**B**) Fluorescence intensity of tetO arrays and centromeric clusters was measured in telophase cells (2 s exposure). Minimum/1st quartile/median/3rd quartile and maximum values are displayed for each group of 50 measurements. Prior to measurement, clusters were separated from lower frequency components of the image (diffuse fluorescence in nuclei and intracellular autofluorescence) by processing the image with wavelet function and adding together scales 1, 2 and 3 (1, 2 and 4 pixels FWHM).

The following figure supplements are available for figure 7:

**Figure supplement 1**. Wavelet filtering allows precise separation of cluster signal from nuclear and cellular background.

Cse4 deposition in anaphase. Taken together, our findings provide compelling evidence that Cse4 is replaced in S phase and remains static on centromeres for the rest of the cell cycle. In this context, budding yeast Cse4 has no epigenetic role in kinetochore inheritance, in contrast to the inheritance of CENP-A on regional centromeres of other organisms (*De Rop et al., 2012*).

The gradual increase in fluorescence intensity observed for Cse4-GFP and Cse4-tdEos after S phase deposition is a manifestation of fluorophore maturation. Accordingly, interpretation of fluorescence intensities for proteins undergoing synthesis and exchange at a highly specific moment of the cell cycle requires caution. Furthermore, in conventional microscopy, centromere clusters frequently appear more point-like in anaphase and telophase than in G1, which may give the impression of a rise in fluorescence when viewed against the increased nuclear background caused by C-terminal Cse4-GFP fusions (*Joglekar et al., 2006*; *Aravamudhan et al., 2013*). A new super-resolution 3D-PALM approach allowed mapping of the actual spatial distribution of individual Cse4 molecules in the centromere cluster, indicating that it should not be treated as a point source for photometric analysis, and providing resolution superior to previous results based on bulk analysis (*Haase et al., 2013*). Moreover, 3D-PALM directly reveals that centromere clusters contract in anaphase. This may be a consequence of the hydrodynamic drag of segregating chromosomes, and is consistent with EM tomography showing congregation of the plus ends of spindle microtubules during anaphase (*O'Toole et al., 1999*). Such compaction of centromere clusters leads to ~threefold higher spatial density of centromeres, increasing the likelihood that individual Cse4 molecules on separate centromeres come into proximity sufficient for FRET. This may explain the higher FRET efficiency reported in anaphase (*Shivaraju et al., 2012*) as interactions between centromeres, without the need to invoke structural oscillation of the centromeric nucleosome between hemisome and octasome.

Previous biochemical and molecular genetic evidence led to a model for a single centromeric nucleosome per yeast chromosome, each containing two Cse4 molecules, located at the ~125 bp *CEN* sequence common to all 16 yeast chromosomes (*Chen et al., 2000*; *Smith, 2002*; *Furuyama and Biggins, 2007*). In contrast, the use of C-terminally tagged Cse4 yielded estimates ranging from 1 to 8 Cse4 molecules per centromere (*Joglekar et al., 2006*; *Coffman et al., 2011*; *Lawrimore et al., 2011*;

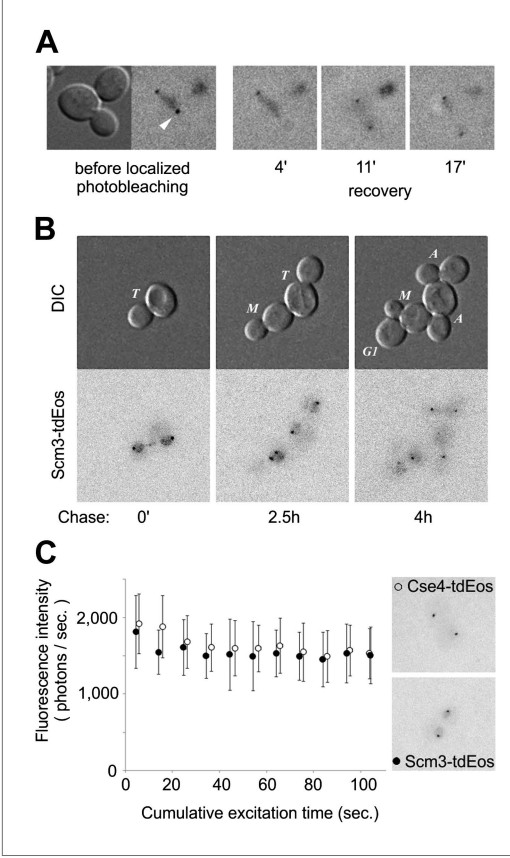

**Figure 8**. Scm3 dynamically interacts with centromeres at levels equivalent to Cse4. (**A**) Scm3-tdEos fluorescence recovery after targeted photobleaching. The experiment was performed essentially as shown in *Figure 5A*, except that recovery was monitored by repetitive imaging, without additional photoconversion. In this example, images were acquired only when indicated. Arrowhead indicates targeted centromere cluster. (**B**) Pulse-chase demonstrates overall stability and cell-cycle persistence of Scm3-tdEos. Photoconverted Scm3 molecules were followed after approximately one and two cell cycles (2.5 and 4 hr, respectively). Fluorescent images are displayed with the same intensity range. (**C**) Fluorescence intensity of centromeric clusters containing Scm3-tdEos (black circles) or Cse4-tdEos[internal] (open circles) in late anaphase/telophase. Average and standard deviation are shown as a function of excitation time to illustrate photostability of photoconverted tdEos. ~10,000 photons were initially detected in 5 s exposure in both cases. Representative images of individual cells containing Cse4-tdEos[internal] or Scm3-tdEos are shown at the same brightness scale.

The following figure supplements are available for figure 8:

**Figure supplement 1**. Scm3-tdEos is present on centromeres and in the nucleus at every stage of the cell cycle.

**Figure supplement 2**. Similar amounts of Cse4 and Scm3 molecules reside at centromeres.

*Shivaraju et al., 2012*; *Aravamudhan et al., 2013*). These discrepant results can be attributed to inaccuracies in estimating spot intensity in the presence of substantial nuclear background, failure to account for the full extent of the centromere cluster (which clearly exceeds the diffraction disk, especially in interphase) in the measurement aperture, or treatment of the cluster as a point source with Gaussian intensity distribution. Interestingly, bimolecular fluorescence complementation (BiFC) experiments demonstrated that the C-terminal Cse4-GFP fusion is deposited on centromeres as a pair during S phase, and the fluorescence intensity of a 'lagging' centromere in a dicentric chromosome at anaphase is consistent with the presence of two Cse4-GFP molecules (*Aravamudhan et al., 2013*). Our photometry measurements of internally tagged Cse4-GFP—taking into account the dimensions of centromere clusters—also support the presence of two molecules of Cse4 in the singular centromeric nucleosome.

The Scm3 chaperone persists at centromeres in every stage of the cell cycle (*Xiao et al., 2011*). This steady-state centromeric occupancy is the result of continuous dynamic exchange, on a timescale of several minutes, with a large nuclear pool of free Scm3 molecules. Such exchange was also observed by *Luconi et al. (2011)* in anaphase, although authors did not reliably observe Scm3 in other stages of the cell cycle. Scm3 may dissociate stochastically, and re-associate onto centromeres through interactions with Ndc10 and AT-rich *CEN* DNA (*Xiao et al., 2011*; *Cho and Harrison, 2011a*). This dynamic property explains the lack of Scm3 in biochemical purifications of kinetochores (*Westermann et al., 2003*; *Akiyoshi et al., 2009*), its absence as a stable component of reconstituted Cse4 octasome (*Dechassa et al., 2011*) and fluctuations in measurements of Scm3 occupancy by ChIP (*Luconi et al., 2011*; *Mishra et al., 2011*; *Shivaraju et al., 2011*; *Xiao et al., 2011*). As a Cse4-specific histone chaperone, Scm3 needs not, in principle, be retained at centromeres once assembly of the centromeric nucleosome has been accomplished in S phase. Indeed, biochemical experiments document classic chaperone properties for the conserved Cse4-binding domain of Scm3 (*Dechassa et al., 2011*; *Shivaraju et al., 2011*; *Xiao et al., 2011*), and NMR and crystal structures of this domain show that DNA binding by Cse4-H4 in the nucleosome is physically incompatible with continued Scm3 interaction (*Zhou et al., 2011*; *Cho and Harrison, 2011b*). However, full-length Scm3 (containing the Ndc10 and DNA binding domains) is enriched at

**Table 1.** Recovery time of centromeric Scm3-tdEos after targeted photobleaching

| Cell cycle stage | Mean recovery time (min) | Standard deviation | Sample size |
|---|---|---|---|
| G1 | 5.2 | 2.5 | 9 |
| S | 4.5 | 1.6 | 4 |
| metaphase | 4.9 | 2.3 | 7 |
| anaphase | 4.7 | 2.0 | 13 |
| telophase | 5.1 | 2.7 | 9 |

Note: After targeted photobleaching of photoconverted Scm3-tdEos centromere clusters with 551 nm dye-laser, cells were imaged with stepwise focus changes within -1 μm to +1 μm Z range (7 steps, 333nm apart, 5 sec. exposure per step). G2 clusters were excluded due to their extended size.

centromeres through all of the cell cycle stages (***Xiao et al., 2011***), consistent with live cell imaging, and genetic studies suggesting the importance of Scm3 after Cse4 deposition (***Camahort et al., 2007***). Steady-state occupancy with dynamic exchange has been described for other chromatin proteins, notably the heterochromatin protein HP1, which functions as a platform for assembling gene silencing complexes (***Cheutin et al., 2003***). Thus, it is highly likely that Scm3 remains after deposition of Cse4 to safeguard the integrity of the singular centromeric nucleosome on each budding yeast chromosome.

A model showing the overall fate of Cse4 in the cell cycle is depicted in ***Figure 9***. In G1, a stable Cse4 nucleosome is maintained by steady-state occupancy of Scm3, which would capture and redeposit Cse4-H4 if any stochastic dissociation occurs. Early in S phase, centromeric nucleosomes are disrupted, leading to removal and degradation of old Cse4, with kinetochore detachment (***Kitamura et al., 2007***). The centromeric nucleosome is then re-established in a step-wise process, most likely starting with binding of CBF1 to CDEI, and CBF3 to CDEIII with assistance of Scm3 (***Camahort et al., 2007***; ***Mizuguchi et al., 2007***). Subsequently, two Scm3-Cse4-H4 heterotrimers are recruited by Ndc10, the dimeric component of CBF3 (***Cho and Harrison, 2011a***). Scm3 then deposits each Cse4-H4 on *CEN* DNA through a dimer intermediate to form a (Cse4-H4)$_2$ tetrasome (***Dechassa et al., 2014***). During this step, CDEII DNA out-competes Cse4-H4 contacts with Scm3, which nonetheless remains in close proximity through interactions with Ndc10 and AT-rich *CEN* DNA. Assembly of two H2A-H2B dimers is likely to follow, although their topography may be altered, as indicated by lack of formaldehyde cross-linking (***Mizuguchi et al., 2007***; ***Xiao et al., 2011***; ***Krassovsky et al., 2012***). Thus, the stable state of Cse4-nucleosomes is octameric, although transient, sub-octameric intermediates may occur during assembly or disassembly. By remaining in close proximity, Scm3 serves not as a structural

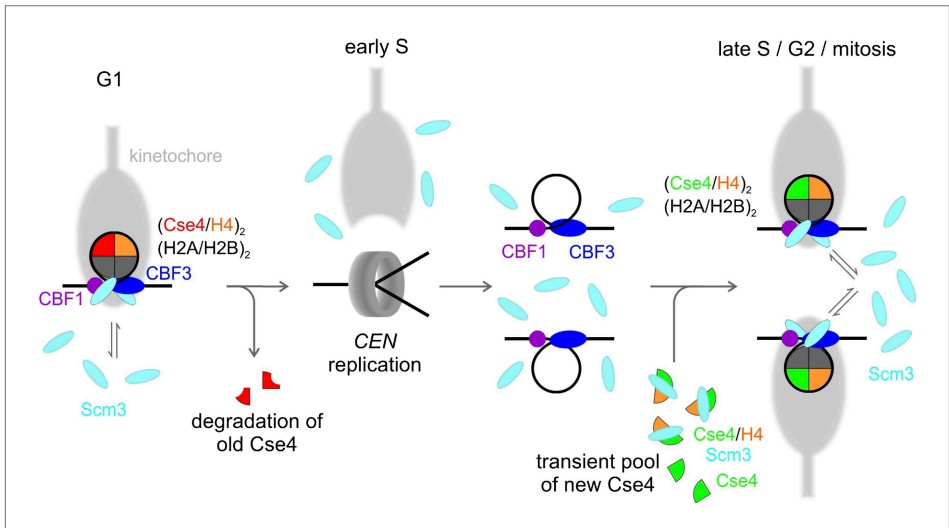

**Figure 9**. Model of Cse4 replacement and re-establishment of point centromere during cell cycle (see text for details).

replacement for H2A-H2B (contrary to our initial model in *Mizuguchi et al. (2007)*), but rather as a persistent chaperone-in-residence to insure against catastrophic loss of the singular Cse4 nucleosome. Given that fungal Scm3 orthologs possess a diversity of DNA binding motifs (*Aravind et al., 2007*), the centromeric persistence of this chaperone through the majority of the *Schizosaccharomyces pombe* cell cycle (*Pidoux et. al, 2009*; *Williams et al., 2009*) or the entirety of the *Saccharomyces cerevisiae* cell cycle (this study) may be a common theme of CENP-A/CenH3 chaperone function. We hope that our findings and clarification of the fates of Cse4 and Scm3 will enable constructive dissection of the mechanisms underlying kinetochore establishment and maintenance to ensure accurate segregation of daughter chromosomes.

# Materials and methods

## Yeast strains

All strains were derived from *Saccharomyces cerevisiae* W1588-4C (*Table 2*) except strain MS173 containing C-terminal Cse4-GFP (MATa his3-1 leu2-0 ura3-0 Cse4-GFP::SpHIS5), which was obtained from Jennifer Gerton, Stowers Institute. Constitutive activity of a mutant URA3 promoter (−80 to −109 deletion; *Roy et al., 1990*) was used for low level expression of TetR-GFP.

## Bacterial protein expression and purification

Histidine-tagged versions of Cse4, Cse4-GFP[internal], and Cse4-tdEos[internal] were expressed in Rosetta (DE3) *Escherichia coli* strain (Novagen, San Diego, CA) under control of T7 promoter (pET15b vector). Bacterial cells were lysed in 6 M guanidine-hydrochloride, 50 mM Tris–HCl pH8, 0.5 mM DTT, 20 mM imidazole buffer, and sonicated. Lysates were absorbed with HisTrap resin (GE Healthcare, Uppsala, Sweden), washed with 8 M urea, 50 mM Tris–HCl pH8, 0.5 mM DTT, 20 mM imidazole buffer and bound protein eluted with the same buffer containing 500 mM imidazole. Concentration of full-length recombinant proteins was assayed by densitometric analysis of Coomassie-stained (Simply Blue Safe Stain, Invitrogen, Carlsbad, CA) SDS-PAGE gel containing known amounts of BSA (fraction V, Sigma-Aldrich, St. Louis, MO).

## Protein analysis

Total cellular extracts were prepared by boiling pelleted yeast samples in SDS loading buffer. After SDS-PAGE, Western blots were probed with affinity-purified rabbit anti-Cse4 (*Mizuguchi et al., 2007*) or anti-H4 antibodies (Upstate, Lake Placid, NY), followed by anti-rabbit IgG-HRP (Life Technologies, Grand

**Table 2.** *Saccharomyces cerevisiae* strains

| Strain | Genotype |
| --- | --- |
| MBY507* | MATa ade2 CSE4-GFP-CSE4 can1-100 his3-11,15 leu2-3,112 trp1-1 ura3-1 RAD5 |
| JBY119† | MATa ADE2 dynLC::hphMX4 cse4::natMX4 can1-100 his3-11,15 leu2-3,112::LEU2-CSE4-tdEOS-CSE4 trp1-1 ura3-1 RAD5 |
| JBY111‡ | MATa ADE2 dynLC::hphMX4 SCM3-tdEOS-kanMX4 can1-100 his3-11,15 leu2-3,112 trp1-1 ura3-1 RAD5 |
| JBY251§ | MATa ADE2 can1-100 his3-11,15 leu2-3,112::LEU2-Δ80ura3p-TetR-GFP-TAP-ADHt trp1-1 ura3-1::pRS406-7xtetO RAD5 |
| JBY252§ | MATa ADE2 can1-100 his3-11,15 leu2-3,112::LEU2-Δ80ura3p-TetR-GFP-TAP-ADHt trp1-1 ura3-1::pRS406-14xtetO RAD5 |
| JBY253§ | MATa ADE2 can1-100 his3-11,15 leu2-3,112::LEU2-Δ80ura3p-TetR-GFP-TAP-ADHt trp1-1 ura3-1::pRS406-21xtetO RAD5 |
| JBY254§ | MATa ADE2 can1-100 his3-11,15 leu2-3,112::LEU2-Δ80ura3p-TetR-GFP-TAP-ADHt trp1-1 ura3-1::pRS306-112xtetO RAD5 |
| MSY173# | MATa his3-1 leu2-0 ura3-0 Cse4-GFP::SpHIS5 |

Notes: *Cse4-GFP[internal], *Xiao et al., 2011*.
†Cse4-tdEos[internal], this paper.
‡Scm3-tdEos[C-terminal], this paper.
§7, 14, 21 or 112 tetO repeats, respectively, and TetR-GFP, this paper.
#Cse4-GFP[C-terminal], *Shivaraju et al., 2012*.

Island, NY). Chemiluminescence was detected with ImageQuant LAS3000 (FujiFilm, Tokyo, Japan). Serial dilutions of recombinant proteins (added to indicated lysates; see *Figure 1—figure supplement 2*) were used to estimate the amounts of endogenous Cse4, Cse4-GFP, and Cse4-tdEos present in yeast lysates after Western blotting with anti-Cse4 antibody.

## Cell cycle stage assignment, synchronization and inhibitor treatment

The bud size and the number/position of centromere clusters were used to assign stages of the cell cycle. For synchronization, low density cultures (OD$_{600}$ <0.3) in CSM medium (MP Biomedicals, Santa Ana, CA) supplemented with 400 µg/ml adenine (Sigma-Aldrich, St. Louis, MO), were exposed to 5 µg/ml of α-factor (Sigma-Aldrich) for 90 min, collected by filtration, washed with sterile water and released into CSM+adenine medium. Entry into S phase (bud emergence, ~45 min after release) was monitored by DIC. Release medium containing 0.2 M hydroxyurea (Sigma-Aldrich) was used to inhibit DNA replication. To block protein synthesis, release medium was supplemented with 200 µg/ml of cycloheximide (Sigma-Aldrich) 10 min after first detection of cells entering S phase (~55 min after release).

## Microscopy

Hamamatsu C9100-13 camera (−94°C, 0.63 MHz, 16-bit ADC; Hamamatsu, Bridgewater, NJ) was used typically with EM gain of 50 (conversion factor 0.044 e$^-$/ADU, readout noise 0.470 e$^-$RMS, thermal current 0.014 e$^-$/s, established experimentally—see *Berry and Burnell, 2011*). IR was blocked with FF01-750/SP filter (Semrock, Rochester, NY). Zeiss AxioObserver Z1 microscope (Carl Zeiss Microscopy, Thornwood, NY) was equipped with Zeiss Plan-Apochromat 150x NA1.35 glycerine-immersion objective, P-737 piezoelectric stage (Physik Instrumente, Auburn, MA), Zeiss Colibri and Lumencor Spectra-6 (Lumencor, Beaverton, OR) illuminators, and custom fluorescence cubes (*Table 3*). Yeast were grown in complete darkness in the CSM+adenine medium (at 25°C, 250 rpm, final OD$_{600}$ ≤0.3), manipulated only under dim red light (660 nm) and imaged in CellAsic Y04C microfluidic chambers (CellASIC, Hayward, CA). 671 nm narrowband illumination (#65-233; Edmund Optics, Barrington, NJ) was used for DIC. To minimize phototoxicity, low level excitation (~7 W/cm$^2$, 1–5 s exposure) was used for fluorescence imaging and 405 nm light (~0.7 W/cm$^2$, 7–10 s) for tdEos photoconversion. Typically, Z-stacks consisted of 13 steps, 333 nm apart.

## Image calibration and display

Raw 16 bit images were converted into FITS format (*Supplementary file 1* contains batch FITS converter macro for ImageJ) and calibrated in 32-bit floating-point space using bias, thermal and flat-field frames (AIP4WIN, *Berry and Burnell, 2011*). Z-stacks were reduced to the composite image only for the presentation purposes, by projecting individual layers, with centromeres in focus, onto a common plane and the identical brightness range was kept for all comparable panels of any given Figure.

## Photometry

All intensity measurements were carried on calibrated, unreduced Z-stacks with aperture photometry in AIP4WIN software, using typical FWHM of centromere cluster (4 pixels = 428 nm) as a radius of measurement aperture and an outer background annulus (5 pixels = 535 nm wide, area 4 times

**Table 3.** Light sources and filters used for wide field fluorescence imaging

| Fluorophore | Light source & channel | Filter cube | | |
| --- | --- | --- | --- | --- |
| | | Excitation | Beamsplitter | Emission |
| GFP | Colibri/LED470* Spectra-6/C§ | FF01-475/28† LL01-488/1† | T495LP‡ | FF01-525/50† |
| tdEos (red emission) | Colibri/LED555* Spectra-6/GY§ | BP550/25* FF01-543/3† | FT570* FF568-Di01† | BP605/70* FF01-593/46† |
| tdEos (photoconversion) | Colibri/LED405* Spectra-6/V§ | FF01-405/10† | 59004BS‡ | 59004M‡ |

Source: *Zeiss.
†Semrock.
‡Chroma.
§Lumencor.

larger than measurement aperture – see *Berry and Burnell, 2011* for discussion of photometry techniques). Background-corrected signal was converted into photoelectrons (equivalent of detected photons) using experimentally established camera parameters (see *Berry and Burnell, 2011* for details).

### Wavelet-based signal extraction

For strains with considerable nuclear background (tetO/TetR-GFP and C-terminal Cse4-GFP strains), the signal corresponding to tetO arrays or centromeric clusters was separated from the diffuse nuclear and cellular background by *à trous* wavelet transform of 32-bit floating point images (see above) using AIP4WIN software (*Berry and Burnell, 2011*). Wavelet scales 1, 2 and 3 were added together to include all objects up to 4 pixels FWHM across and the intensity of spots was measured by aperture photometry as above.

### Targeted photobleaching

Galvano-controlled MicroPoint system (Photonic Instruments, Saint Charles, IL) was used for targeted photobleaching. 551 nm pulsed dye laser was focused to a diffraction-limited spot (FWHM ~210 nm) and centromeres were targeted in real-time during initial Z-stack acquisition, after photoconversion. Following recovery, additional Z-stacks were acquired from time to time.

### MFM-PALM

Multifocus microscope (*Abrahamsson et al., 2013*) was used for 3D-PALM. The system contained MFM grating (designed to yield 380 nm spacing between consecutive planes in the multifocal image), matching corrective grating/prism, Nikon 100x NA1.4 oil-immersion objective and Andor DU897+ camera (−70°C, EM gain = 250, 70 ms exposure; Andor Technology USA, South Windsor, CT), yielding final image voxel of 120 × 120 × 380 nm. Paraformaldehyde-fixed yeast cells were attached to conca-navalin A-coated cover slips containing immobilized 550 nm bare Gold Nanorods (25 nm diameter, 550 nm emission; NanoPartz, Loveland, CO; *Shtengel et al., 2009*). For limited photoconversion, a 405 nm laser (Coherent, Santa Clara, CA) was used at 0.2W/cm$^2$, and red tdEos fluorophores were detected under 561 nm laser illumination (2 kW/cm$^2$; Cobolt, San Jose, CA). Images were corrected for distortion and transmission, converted into 3D stacks, then individual events were identified and their 3D coordinates determined with FISHQuant (*Mueller et al., 2013*). Finally, residual 3D drift of the sample was corrected in MatLab (MathWorks, Natick, MA) based on Gold Nanorod fiducials. ViSP software (*El Beheiry and Dahan, 2013*) was used for visualization and presentation of results. Full details for MFM-PALM are available upon request (Hajj et al., unpublished data).

## Acknowledgements

We thank Manjunatha Shivaraju and Jennifer Gerton (Stowers Institute for Medical Research, Kansas City) for the C-terminally tagged Cse4-GFP strain, Gleb Shtengel (Janelia Farm, Ashburn) for advice and preparation of coverslips with immobilized Gold Nanorods, Richard Baker for communicating unpublished results, and Mohamed El Beheiry for use of ViSP software during its development. We are grateful to members of Transcription Imaging Consortium (HHMI Janelia Farm, Ashburn) and our laboratories for helpful discussions. This work was supported by the Center for Cancer Research, NCI, NIH, and the Janelia Farm Research Campus, HHMI. MD acknowledges the support of a Fulbright fellowship.

## Additional information

### Funding

| Funder | Author |
|---|---|
| Howard Hughes Medical Institute (HHMI) | Jan Wisniewski, Bassam Hajj, Jiji Chen, Gaku Mizuguchi, Maxime Dahan, Carl Wu |
| National Institutes of Health (NIH) | Jan Wisniewski, Gaku Mizuguchi, Hua Xiao, Debbie Wei, Carl Wu |

The funders had no role in study design, data collection and interpretation, or the decision to submit the work for publication.

## Author contributions

JW, Conception and design, Acquisition of data, Analysis and interpretation of data, Drafting or revising the article, Contributed unpublished essential data or reagents; BH, JC, HX, Acquisition of data, Analysis and interpretation of data, Drafting or revising the article; GM, Conception and design, Drafting or revising the article, Contributed unpublished essential data or reagents; DW, Acquisition of data, Analysis and interpretation of data; MD, Analysis and interpretation of data, Drafting or revising the article; CW, Conception and design, Analysis and interpretation of data, Drafting or revising the article

## Additional files

### Supplementary file

• Supplementary file 1. **Batch FITS converter macro for ImageJ**. This macro converts 16-bit TIFF files from a selected folder into FITS files (re-assigned into 32-bit floating-point space) and saves them in a destination folder. The content of the file should be saved as 'Batch FITS Converter.txt' into Macro folder of ImageJ.

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
