## [Decision Letter]

Thank you for sending your work entitled “Imaging Cse4 histone fate reveals de novo replacement in S phase and subsequent stable residence at centromeres” for consideration at *eLife*. Your article has been favorably evaluated by a Senior editor, Jim Manley, and 3 reviewers, one of whom, Jim Kadonaga, is a member of our Board of Reviewing Editors.

The Reviewing editor and the other reviewers discussed their comments before we reached this decision, and the Reviewing editor has assembled the following comments to help you prepare a revised submission.

The budding yeast centromeric nucleosome has been the subject of many studies from many laboratories, but there has been considerable disagreement with regard to this essential part of the centromere in an organism that has been invaluable for centromere studies. This paper by Wisniewski et al. on Cse4 (CenH3) represents an important step forward in our understanding of the yeast centromere. In this study, the authors carried out fluorescence pulse-chase experiments with internally-labeled Cse4 and found that preexisting Cse4 is replaced with newly-synthesized Cse4 during S phase in a DNA replication-dependent process. In general, the experiments were carefully performed, and the work has been presented in a lucid and logical manner.

The reviewers feel that this work is generally appropriate for publication for *eLife*. There is, however, one important issue that needs to be addressed by the authors. As seen in Figure 1, the levels of internally-tagged Cse4 (lanes 2 and 3) appear to be roughly two-fold lower than the level of wild-type Cse4 (lane 1). The authors should address whether or not the apparently lower levels of internally-tagged Cse4 are due to a technical issue (e.g., inefficient transfer to the membrane) or to genuinely lower levels of the tagged proteins. If the latter, the lower levels of tagged proteins need to be incorporated into the interpretation of the results. To test if it is the former, the authors might try a quantitative method such as SILAC with wild-type vs. internally-tagged strains.

In other words, the stoichiometry within the nucleosome and the number of nucleosomes have been contentious issues. Moreover, the blot (Figure 1), taken at face value, says that the number of Cse4 molecules in their internally-tagged versions is substantially less than in the WT strain. Hence, this is an important point because assessing the amount of Cse4 at centromeres throughout the cell cycle and counting the number of Cse4 molecules are two of the main goals in this study.

Minor points:

1) As the authors correctly point out, the accurate counting of molecules, as distinguished from counting fluorescent emissions or “events”, is very difficult by localization microscopy. The authors emphasize the sources of error in under-counting molecules. It is probably worth noting that counting is difficult because of both under- and over-counting errors. The major over-counting error comes from the fact that a single molecule of mEOS (and I presume tdEOS) can blink multiple times, each registering as an independent event (see for example [36] PMID: 23045631).

2) We are slightly uncomfortable with the apparent precision of the photometrics results arguing for a maximum of 2 Cse4 molecules per centromere because the TetR-GFP images have such a high nuclear background and thus require major image processing to extract the array signal. This background is not unique to this work and is consistent with other reports. In retrospect, this problem might have been mitigated by using a lacO and LacI-GFP combination. Nevertheless, the results are carefully documented, consistent, and reasonable, and my concern is minor.

3) In the first paragraph of the Results section, there is a problem with the writing that causes it to sound illogical. The authors’ mention that the C-terminus is 'almost buried' and that is a reason for not tagging there, but then that this is an important surface for Mif2 binding. “Almost buried” is not an appropriate description of this region, since it is clearly surface-exposed. I think the point could be made that the C-terminus of Cse4 is very close to the structured part of the nucleosome and that it also serves as an important binding surface. Both of those reasons make the functional problems seen in the C-term fusions seem logical, which is what I think the authors were trying to get at.

4) In the Results section, can the authors clarify the 'free fluorescent moieties'. Why do we think these exist? Why do we think we know how they come about?

5) The Abstract states that their “findings indicate that a stable Cse4 nucleosome is maintained by chaperone-in-residence Scm3 throughout the cell cycle”, but this manuscript appears to lack data that directly demonstrate this conclusion. This point might be better made in the Discussion rather than in the Abstract.

---

## [Author Response]

*The reviewers feel that this work is generally appropriate for publication for eLife. There is, however, one important issue that needs to be addressed by the authors. As seen in*
Figure 1*, the levels of internally-tagged Cse4 (lanes 2 and 3) appear to be roughly two-fold lower than the level of wild-type Cse4 (lane 1). The authors should address whether or not the apparently lower levels of internally-tagged Cse4 are due to a technical issue (e.g. inefficient transfer to the membrane) or to genuinely lower levels of the tagged proteins. If the latter, the lower levels of tagged proteins need to be incorporated into the interpretation of the results. To test if it is the former, the authors might try a quantitative method such as SILAC with wild-type vs. internally-tagged strains*.

*In other words, the stoichiometry within the nucleosome and the number of nucleosomes have been contentious issues. Moreover, the blot (*Figure 1*), taken at face value, says that the number of Cse4 molecules in their internally-tagged versions is substantially less than in the WT strain. Hence, this is an important point because assessing the amount of Cse4 at centromeres throughout the cell cycle and counting the number of Cse4 molecules are two of the main goals in this study.*

To address the apparently low level of internally tagged Cse4, we have constructed bacterial strains expressing His6-Cse4, His6-Cse4-GFPinternal and His6-Cse4-tdEos, purified the recombinant proteins by Ni-Sepharose chromatography, and determined protein concentration. We performed quantitative Western blotting by spiking known amounts (fmoles) of purified proteins into yeast extracts and used the Western blot signals to calculate the amount of the corresponding endogenous proteins. As noted in the revised text and shown in Figure 1—figure supplement 2, the levels of Cse4-GFPinternal and Cse4-tdEosinternal are comparable to WT Cse4. This suggests that the lower signal of internally tagged Cse4 in Figure 1—figure supplement 2 is indeed due to lower blotting efficiency of the substantially larger, tagged proteins. We recognize that the use of SILAC would provide greater precision, but a protein of such low cellular abundance as Cse4 presents a logistical challenge for future consideration.

Minor points:

*1) As the authors correctly point out, the accurate counting of molecules, as distinguished from counting fluorescent emissions or “events”, is very difficult by localization microscopy. The authors emphasize the sources of error in under-counting molecules. It is probably worth noting that counting is difficult because of both under- and over-counting errors. The major over-counting error comes from the fact that a single molecule of mEOS (and I presume tdEOS) can blink multiple times, each registering as an independent event (see for example*
[36]
*PMID: 23045631)*.

To address the apparently low level of internally tagged Cse4, we have constructed bacterial strains expressing His6-Cse4, His6-Cse4-GFPinternal and His6-Cse4-tdEos, purified the recombinant proteins by Ni-Sepharose chromatography, and determined protein concentration. We performed quantitative Western blotting by spiking known amounts (fmoles) of purified proteins into yeast extracts and used the Western blot signals to calculate the amount of the corresponding endogenous proteins.

As noted in the revised text and shown in Figure 1—figure supplement 2, the levels of Cse4-GFPinternal and Cse4-tdEosinternal are comparable to WT Cse4. This suggests that the lower signal of internally tagged Cse4 in Figure 1—figure supplement 2 is indeed due to lower blotting efficiency of the substantially larger, tagged proteins. We recognize that the use of SILAC would provide greater precision, but a protein of such low cellular abundance as Cse4 presents a logistical challenge for future consideration.

*2) We are slightly uncomfortable with the apparent precision of the photometrics results arguing for a maximum of 2 Cse4 molecules per centromere because the TetR-GFP images have such a high nuclear background and thus require major image processing to extract the array signal. This background is not unique to this work and is consistent with other reports. In retrospect, this problem might have been mitigated by using a lacO and LacI-GFP combination. Nevertheless, the results are carefully documented, consistent, and reasonable, and my concern is minor*.

We agree that the high nuclear background of TetR-GFP complicates intensity measurements. To reduce background, we have now expressed TetR-GFP under control of a mutant ura3 promoter under noninducing conditions. This gives comparable photometric values as before, and because the background is now lower, it is also visually apparent that the intensity values for Cse4-GFPinternal falls between the TetR-GFP signals for 14x and 21x TetO. (Figure 7 and Figure 7—figure supplement 1).

*3) In the first paragraph of the Results section, there is a problem with the writing that causes it to sound illogical. The authors’ mention that the C-terminus is 'almost buried' and that is a reason for not tagging there, but then that this is an important surface for Mif2 binding. “Almost buried” is not an appropriate description of this region, since it is clearly surface-exposed. I think the point could be made that the C-terminus of Cse4 is very close to the structured part of the nucleosome and that it also serves as an important binding surface. Both of those reasons make the functional problems seen in the C-term fusions seem logical, which is what I think the authors were trying to get at*.

We thank reviewers for helping clarify the point we wish to make. This has been introduced in the revised text.

*4.) In the Results section, can the authors clarify the 'free fluorescent moieties'*. *Why do we think these exist? Why do we think we know how they come about?*

We have revised the text to clarify the issue of cytoplasmic events. We base our explanation on published work (referenced in text) concerning regulation of Cse4 levels by proteolysis and resistance of GFP-type proteins to degradation. Our explanation is that cytoplasmic detections may correspond to the remnants of proteolytic processing of tagged Cse4 (in contrast to a free pool of Cse4-tdEos, unlikely in the light of pulse-chase results). However, we cannot exclude other sources of fluorescence, so the explanation provided is stated as speculative.

*5) The Abstract states that their “findings indicate that a stable Cse4 nucleosome is maintained by chaperone-in-residence Scm3 throughout the cell cycle”, but this manuscript appears to lack data that directly demonstrate this conclusion. This point might be better made in the Discussion rather than in the Abstract*.

The statement was not intended to be declarative, and we thank reviewers for pointing this out. Accordingly, we have softened the sentence by substituting “indicate” with “suggest”. We do think the point deserves mention in the abstract, as it conveys an important concept that distinguishes Scm3 from conventional histone chaperones that do not persist at the site of histone deposition. This is articulated further in the Discussion.